# Obesity in young South African women living with HIV: A cross-sectional analysis of risk factors for cardiovascular disease

**Sherika Hanley**[1]*, **Dhayendre Moodley**[2], **Mergan Naidoo**[3]

**1** Umlazi Clinical Research Unit, Centre for the AIDS Programme of Research of South Africa, University of KwaZulu-Natal, Durban, South Africa, **2** Department of Obstetrics and Gynaecology, University of KwaZulu-Natal, Durban, South Africa, **3** Department of Family Medicine, University of KwaZulu-Natal, Durban, South Africa

* hanley@ukzn.ac.za, Sherika.hanley@caprisa.org

## Abstract

### Background

Young South African women are faced with a dual epidemic of HIV and obesity, placing them at a high risk of developing atherosclerotic cardiovascular disease (CVD). We sought to determine the prevalence of CVD risk factors in a cohort of reproductive-aged South African women living with HIV (WLHIV).

### Methods

While the main purpose of an ongoing intervention study is the reduction of cardiovascular disease through the integration of CVD screening and prevention in the HIV management plan for women of reproductive age (ISCHeMiA trial), we present the prevalence of risk factors for CVD in this cohort of young women at baseline. Sociodemographic, conventional CVD risk factors, HIV-related factors and self body image perception were assessed through study questionnaires and standardized clinical and laboratory procedures.

### Results

Of the 372 WLHIV enrolled from November 2018 to May 2019, 97% had received efavirenz-based antiretroviral treatment (ART) for at least 1 year and 67.5% (248/367) of women were overweight or obese at the time of enrolment. The prevalence of metabolic syndrome was 17.6% (95%CI 11.6–22.8) at a median age of 35 years (IQR 30.5–40.5). A significant proportion of women had abnormally low levels of high-density lipoprotein (43.2%, 80/185) and elevated levels of high sensitivity C-reactive protein (59.5%, 110/185). Seventy five percent of overweight women with an increased waist circumference reported to be satisfied with their body image.

### Conclusions

The high prevalence of metabolic syndrome, obesity and elevated markers of inflammation in young South African WLHIV, underscores the need for a proactive integrated

**Data Availability Statement:** All relevant data are within the paper and its Supporting Information files.

**Funding:** This work is based on the research supported in part by the National Research Foundation of South Africa (Grant Number: 117730) awarded to SH. https://www.nrf.ac.za/ Research reported in this publication is also supported by the Fogarty International Center (FIC), NIH Common Fund, Office of Strategic Coordination, Office of the Director (OD/OSC/CF/NIH), Office of AIDS Research, Office of the Director (OAR/NIH), National Institute of Mental Health (NIMH/NIH) of the National Institutes of Health under Award Number D43TW010131. SH is a sub awardee. https://www.fic.nih.gov/Funding/Pages/Fogarty-Funding-Opps.aspx The PROMOTE study is funded by the President's Emergency Plan for AIDS Relief (PEPFAR) through DAIDS/NIAID/NIH grants to CAPRISA Clinical Trials Unit for AIDS/Tuberculosis Prevention and Treatment, grant # 5UM1AI069469. SH is a site investigator in the PROMOTE study. https://www.hiv.gov/federal-response/pepfar-global-aids/pepfar The content is solely the responsibility of the authors and does not necessarily represent the official views of the National Research Foundation of South Africa and National Institutes of Health. The funders had no role in study design, data collection and analysis, decision to publish, or preparation of the manuscript.

**Competing interests:** The authors do not have financial, consultant, institutional or other relationships that might lead to a bias or conflict of interest.

management approach to prevent atherosclerotic cardiovascular disease in low and middle income settings.

## Introduction

Globally 19.2 million women and girls are living with HIV, of which 51% reside in South Africa (SA) [1]. Large scale efforts have led to greatly improved access to antiretroviral therapy (ART) in women living with HIV (WLHIV) and a rapid decline in AIDS-related mortality [2]. Conversely, the prevalence of metabolic syndrome and atherosclerotic cardiovascular disease (CVD) has increased in SA and other low and middle income countries (LMICs) as a result of epidemiologic transition and a related increase in conventional risk factors, coupled with HIV and treatment thereof [3–6]. Worldwide the leading cause of death and significant disability is from CVD with highest burden in LMICs highlighting the need for an integrated approach to the prevention and management of chronic communicable and non-communicable diseases (NCDs) [7, 8].

A targeted integrated programme needs to be driven by relevant local evidence of burden of disease however, data on prevalence and incidence of NCDs in WLHIV globally is limited. HIV appears to increase the risk of CVD in women more than it does in men [9]. Cardiovascular disease risk is exacerbated by HIV through direct mechanisms of persistent immune activation and chronic inflammation [10]. Immune activation and inflammation have been shown to be more pronounced in WLHIV than in men, possibly related to sex hormone differences [11]. Women in SA are not only more likely to be affected by the HIV epidemic than their male counterparts but are more prone to develop obesity and the metabolic syndrome compared to men [12]. The obesity epidemic may be further complicated by a self-preference for a higher body mass index in sub-Saharan African women [13].

In this baseline analysis of an ongoing interventional study, we describe HIV-related factors and conventional risk factors for cardiovascular disease in treatment-experienced young South African WLHIV.

## Methods

### Study design

The ISCHeMiA study (**I**ntegration of cardiovascular disease **SC**reening and prevention in the **H**IV **MA**nagement plan for women of reproductive age), is a prospective, quasi-experimental design comparing a primary healthcare intervention plan guided by the WHO Package of Essential Non-communicable Disease interventions for primary health care in low resource settings (WHO PEN) [14] (Intervention Arm), with routine care (Control Arm). Cross-sectional analyses were performed of baseline CVD risk factors in total study population and author-recommended CVD risk factors in the intervention arm. Results of the ongoing intervention are expected to be released in 2022.

### Study population

Women aged between 18 and 49 years, who were receiving ART for a minimum of one year from peri-urban primary health care (PHC) clinics in Umlazi and surrounding rural areas who intended to reside within the study catchment area for the three-year duration of the study, were included in the study. Women in the intervention arm were recruited from a

cohort of participants co-enrolled in the PEPFAR PROMise Ongoing Treatment Evaluation (PROMOTE) observational study at the Umlazi Clinical Research Site. The PROMOTE study has been implemented to provide long-term follow-up data on safety outcomes of use of combination ART received from standard of care healthcare providers. All 238 women in the PRMOTE study are living with HIV, were between 18–49 years The first 186 interested eligible candidates presenting to the research clinic for their next PROMOTE study visit were co-enrolled into the intervention arm and assessed for CVD risk.

For the control arm, the Tier.Net HIV electronic register was used to select a data base of all women with HIV, aged between 18–49 years and receiving ART for more than 1 year, at the nearest Umlazi Gateway PHC. Scheduled clinic visits at similar time points to the anticipated clinic visits in the intervention group were used to establish a list of potentially eligible women. Following a matched pool of data, the first 186 women fulfilling the inclusion criteria, who attended the clinic for their next appointment and who consent to study participation were enrolled.

## Enrolment procedures and data collection at baseline

Written informed consent forms were signed by interested participants in both arms. In the control arm, routinely documented data were extracted from participant medical records. These included demographic characteristics, most recent HIV viral load, CD4 count, ART regimen, duration of ART, height, weight, and blood pressure recordings where available. In the intervention arm, baseline sociodemographic factors, conventional CVD risk factors, HIV-related risk factors, and current self body image acceptance were determined through study questionnaires, physical examination, and laboratory investigations.

**Questionnaires and physical examination.** Questionnaires were guided by the WHO STEPwise approach to NCD risk factor surveillance (STEPS) and tailored to determining CVD risk using the core sections [15]. The WHO Steps questionnaire diet was categorised as unhealthy or healthy based on response to fruit and vegetable intake (high/ low), high fat and high salt diet. Self-perception of diet (unhealthy or healthy) was also included. Duration of exercise per week ($>$/$<$ 30 minutes per week) was recorded.

Height, weight, and blood pressure (BP) measurements were conducted using standard procedures. Waist circumference (WC) was measured by snugly placing a measuring tape in a horizontal plane around the abdomen immediately above the iliac crest at the level of the umbilicus, at the end of expiration. Body mass index (BMI) calculated by weight divided by height squared was classified based on WHO guidelines: overweight (25-$<$30kg/m$^2$), class 1 obesity (30-$<$35 kg/m$^2$), class 2 obesity (35-$<$40 kg/m$^2$) and class 3 obesity ($>$ 40 kg/m$^2$). Hypertension was defined as systolic blood pressure (SBP) $\geq$140mmHg and/or diastolic blood pressure (DBP) $\geq$90 mmHg or receiving antihypertensive treatment. Prehypertension was defined as SBP $\geq$130-139mmHg and/or DBP $\geq$85-89mmHg.

## Laboratory investigations

Laboratory investigations were carried out on overnight fasting samples at two certified laboratories with regular system calibration of validated devices and quality measures in place. High sensitivity CRP was measured via Beckman Coulter AU analyser with detection limits of 0.2–160 mg/L. Per hsCRP latex package insert, CVD relative risk is considered as follows: low $<$ 1mg/L, average 1-3mg/L and high $>$ 3mg/L. Urine microalbumin of $<$3 mg/mmol is considered normal/mildly increased, 3-29mg/mmol is moderately increased, and $>$30mg/mmol is severely increased. Fasting glucose levels of $<$ 5.6, $\geq$ 5.6, 6–6.9, $\geq$ 7 mmol/L are normal, meets metabolic syndrome criteria, impaired fasting glucose and diagnostic of diabetes

mellitus, respectively. Levels in mmol/L of serum total cholesterol (TC) > 5, HDL < 1.2, LDL > 3 and triglyceride > 1.7 are abnormal in adult females.

HIV-1 viral load was measured using COBAS AmpliPrep/COBAS TaqMan HIV-1 test. For the purposes of this study, participants with viral load <200 copies/ml (cp/ml) were considered as virally suppressed. CD4 count was determined by the Becton Dickinson Facscalibur flow cytometer.

**Metabolic syndrome and CVD risk assessments.** Metabolic syndrome (MetS) was defined by the 2009 Joint Interim Statement (JIS) recommended by the South African Heart Association (SA Heart) and the Lipid and Atherosclerosis Society of Southern Africa (LASSA) and includes the following subcomponents: WC ≥ 80 cm, elevated triglycerides ≥ 1.7 mmol/L, reduced HDL <1.3 mmol/L, elevated SBP ≥ 130 and/or DBP ≥ 85 mmHg or on antihypertensive treatment, and an elevated fasting glucose ≥ 5.6 mmol/L or receiving treatment for diabetes mellitus [16].

Baseline CVD risk assessments were calculated using the Framingham (Fr) 5 and 10 year CVD risk [17], WHO and International Society of Hypertension (WHO/ISH) cardiovascular risk prediction [18], and the Data collection on Adverse Effects of Anti-HIV Drugs Study (DAD) coronary heart disease (CHD) equation [19].

## Statistical analysis

Data was captured on Microsoft Excel and analysed using IBM SPSS statistics software, version 25.0 and Epi Info version 7.2.3.1. Conventional CVD risk factors and HIV-related risk factors were represented by means (standard deviation), medians (interquartile ranges) and prevalence. 95% confidence intervals were derived from using Epiinfo 7.0 and were either calculated as a Wilson interval or an Exact interval where appropriate. Wilson 95% Confidence Interval was calculated for Systolic BP, CD4, Age, BMI, ART and ART Duration Groups. Exact 95% Confidence Interval was calculated for all other categories. The relationship between body image perception and BMI was evaluated by Pearson's chi-square test.

## Ethical considerations and approvals

The ISCHeMiA study was conducted in accordance with the ethical standards of University of KwaZulu-Natal (UKZN) Biomedical Research Ethics Committee and with the Helsinki Declaration (1964, amended in 2008). Approvals were obtained from the PEPFAR PROMOTE publication committee, Department of Health KZN and Prince Mshiyeni Memorial Hospital. The trial is registered with Pan African Clinical Trial Registry database, (www.pactr.org), [identification number PACTR201808524461224].

## Results

A total of 372 WLHIV (186 in each of the intervention and control arms) were enrolled from November 2018 to May 2019 for whom data were available for a baseline analysis. Demographic, HIV-related and conventional CVD-related data were collected at enrolment and are displayed in Table 1. Data on age, BMI, systolic blood pressure, antiretroviral drug (ARV) regimen and duration, HIV viral load and CD4 count are included for all women enrolled.

More than 80% of the study population were younger than 40 years when enrolled. Two-thirds (67.5%) of the study population had a BMI > 25kg/m$^2$ and were classified as being overweight or obese. Elevated SBP and DBP were noted in 14.4% and 16.3% of the study population respectively. The majority of women were on an EFV-based first line ART for more than a year (97.3%), virally suppressed (95.1%), and with a CD4 count > 500 cells/IU (81.7%).

**Table 1. Prevalence of CVD risk factors determined as per standard of care in total study population (N = 372).**

|  | Categories | N (%; 95%CI) |
|---|---|---|
| ***Demographic factors*** | | |
| Mean age in years(SD) | 33.5 (6.1) | |
| Age groups (years) N = 372 | 20–29 | 113 (30.4; 25.8–35.4) |
| | 30–39 | 194 (52.2; 46.9–57.3) |
| | 40–49 | 65 (17.5; 13.8–21.8) |
| ***Traditional CVD risk factors*** | | |
| BMI Median (IQR) | 27.3 (23.2–33.1) | |
| BMI categories (kg/m$^2$) N = 367 | <25 | 139 (37.9; 32.9–43.1) |
| | 25–29 | 95 (25.9; 21.5–30.7) |
| | 30–39 | 133 (36.2; 31.4–41.4) |
| | 40–49 | 20 (5.5; 3.5–8.4) |
| Systolic blood pressure Mean (SD) | 115.7 (15.6) | |
| Systolic blood pressure categories (mmHg) N = 368 | <130 | 315 (85.6; 81.5–88.9) |
| | 130–139 | 27 (7.3; 4.9–10.6) |
| | ≥140 | 26 (7.1; 4.8–10.3) |
| Diastolic blood pressure Mean (SD) | 73.3 (11.5) | |
| Diastolic blood pressure categories (mmHg) N = 202 | <85 | 169 (83.7; 77.8–88.5) |
| | 85–89 | 15 (7.4; 4.2–11.9) |
| | ≥90 | 18 (8.9; 5.4–13.7) |
| ***HIV-related factors*** | | |
| ART Regimen (type) N = 372 | AZT/3TC+ LPV/r | 4 (1.1; 0.4–2.9) |
| | EFV/FTC/TDF | 362 (97.3; 94.9–98.6) |
| | FTC/TDF + LPV/r | 5 (1.3; 0.5–3.3) |
| | NVP+FTC/TDF | 1 (0.3; 0.01–1.7) |
| Duration of Current ART Regimen (years) Mean(SD) | 4.3 (2.2) | |
| Duration of Current ART Regimen (years) Categories | 1–4 | 270 (72.6; 67.7–76.9) |
| | >4 | 102 (27.4; 23.0–32.3) |
| CD4 Count (cells/iu) Median (IQR) | 803 (558–1030) | |
| CD4 count (cells/iu) Categories N = 371 | <500 | 68 (18.3; 14.6–22.7) |
| | ≥500 | 303 (81.7; 77.3–85.4) |
| Viral Load (cp/ml) Mean(Range) | 768 (0–113 559) | |
| Viral load (cp/ml) Categories N = 368 | ≥200 | 18 (4.9; 3.0–7.8) |
| | <200 | 350 (95.1; 92.2–96.9) |

Key: SD = Standard deviation, IQR = Interquartile range, CI = confidence interval, AZT = Zidovudine,

3TC = lamivudine, EFV = Efavirenz, FTC = Emtricitabine, TDF = Tenofovir disoproxil fumarate,

LPV/r = Lopinavir/Ritonavir, NVP = Nevirapine.

Additional data required by most standard tools for assessing CVD risk are listed in Table 2. Assessing these additional risk factors constitute the WHO PEN intervention package that is applied to the intervention arm only in the ISCHeMiA trial. Review of medical history of the women in the intervention arm revealed a 10.2% prevalence of systemic hypertension and 0.5% Type 2 Diabetes Mellitus (T2DM).

Apart from the woman with known T2DM, screening of fasting blood glucose identified one other woman (0.5%) with impaired fasting glucose. Three women (1.6%; 95%CI 0.3–4.7) had fasting glucose levels > 5.6 mmol/L, a subcomponent of MetS. Almost forty five percent (82/183) of women in the intervention arm were obese (≥30kg/m$^2$); and 79/80 (98.8%) of

**Table 2.** Prevalence of additional CVD risk factors investigated in the intervention arm (N = 186).

| | Categories | N (%; 95%CI) |
|---|---|---|
| *Sociodemographic factors* | | |
| Employment N = 186 | Yes | 94 (50.5; 43.1–57.9) |
| | No | 92 (49.5; 42.1–56.9) |
| Parity Median (IQR) | 2 (1–3) | |
| Parity N = 186 | 0–2 | 55 (30.4; 23.8–37.7) |
| | >2 | 88 (47.3; 39.9–54.8) |
| *Traditional CVD risk factors-non modifiable* | | |
| Family History of CVD | Yes | 18 (9.7; 5.8–14.9) |
| | No | 168 (90.3; 85.1–94.2) |
| *Modifiable CVD risk factors* | | |
| Known with Hypertension N = 186 | No | 167 (89.8; 84.5–93.7) |
| | Yes | 19 (10.2; 6.3–15.5) |
| Known with Diabetes N = 186 | No | 185 (99.5; 97.0–99.9) |
| | Yes | 1 (0.5; 0–2.9) |
| Fasting Plasma Glucose mmol/L Mean (SD) | 4.5 (0.4) | |
| Glucose N = 185 | <5.6 | 181 (97.8; 94.6–99.4) |
| | 5.6–5.9 | 3 (1.6; 0.3–4.7) |
| | 6–6.9 | 1 (0.01–2.9) |
| | ≥7 | 0 |
| Waist Circumference cm Mean (SD) | 89.2 (14.9) | |
| Waist Circumference Categories N = 181 | <80 | 55 (30.4; 23.8–37.7) |
| | ≥80 | 126 (69.6; 62.3–76.2) |
| Total Cholesterol mmol/L Mean(SD) | 4.0 (0.8) | |
| Total Cholesterol N = 185 | <5 | 167 (90.3; 85.1–94.1) |
| | ≥5 | 18 (9.7; 5.9–14.9) |
| HDL mmol/L Mean(SD) | 1.3 (0.3) | |
| HDL mmol/L N = 185 | <1.2 | 80 (43.2%; 35.9–50.7) |
| | ≥1.2 | 105 (56.8%; 49.3–64.0) |
| Triglyceride mmol/L Median (IQR) | 0.72 (0.57–1.00) | |
| Triglyceride mmol N = 185 | <1.7 | 174 (94.1; 89.6–96.9) |
| | ≥1.7 | 11 (5.9; 3.0–10.4) |
| LDL mmol/L Mean(SD) | 2.3 (0.8) | |
| LDL mmol/L = 186 | <3 | 153 (82.3; 76.0–87.5) |
| | ≥3 | 33 (17.7; 12.5–24.0) |
| hsCRP mg/L Median (IQR) | 3.7 (1.4–9.4) | |
| hsCRP mg/L N = 185 | <3 | 75 (40.5; 33.4–47.9) |
| | ≥3 | 110 (59.5; 52.0–66.6) |
| Urine Albumin/Creatinine Ratio mg/mmol Median (IQR) | 0.5 (0.35–0.90) | |
| Urine Albumin/Creatinine ratio mg/mmol N = 184 | <3 | 25 (13.6; 8.9–19.4) |
| | ≥3 | 159 (86.4; 80.6–91.1) |
| Unhealthy Diet | Yes | 91 (49.7; 42.8–57.8) |
| | No | 90 (50.3; 42.2–57.4) |
| Exercise mins/wk Mean (SD) | 42 (74.3) | |
| Exercise (>30 mins/wk) | Yes | 64 (34.4; 27.6–41.7) |
| | No | 122 (65.6; 58.3–72.4) |
| Alcohol Consumption | Yes | 44 (23.7; 17.8–30.4) |
| | No | 142 (76.3; 69.6–82.3) |

(*Continued*)

**Table 2.** (Continued)

|  | Categories | N (%; 95%CI) |
|---|---|---|
| Ever Smoked | Yes | 18 (9.7; 5.6–14.9) |
|  | No | 168 (90.3; 85.1–94.2) |

Key: SD = Standard deviation, IQR = Interquartile range, CI = confidence interval.

obese women had a high WC (>80 cm). Overall, 79/183 (43.2%) women in the intervention arm had both high BMI >30kg/m$^2$ and a high WC.

Other components of MetS noted in participants at baseline included an elevated fasting TC (9.7%), low HDL (43.2%), elevated level of LDL (17.7%), and elevated level of triglycerides (5.9%). A significant proportion of women had elevated inflammatory markers, hsCRP (59.5%). Decreased urine albumin-creatinine ratio was noted in 13.6% of women.

Lifestyle practice is also demonstrated in Table 2 with almost 50% of the women in the intervention arm reporting an unhealthy diet, and significantly less (34.4%) women reported to exercise more than 30 minutes per week.

### Prevalence of Metabolic Syndrome (MetS)

The prevalence of metabolic syndrome at baseline in the intervention arm was 17.6% (95%CI 11.6–22.8) based on meeting 3 or more subcomponents per Joint Interim Statement (JIS). Seven combinations of subcomponents are displayed in (Fig 1). Of the 31 women with MetS, the most common combination at 61% (19/31) was elevated SBP≥ 130 mmHg and/or DBP≥85 mmHg, WC ≥80 cm and reduced HDL <1.3 mmol/L. Elevated WC was present across all women with Mets at a mean WC of 91.2cm. Elevated HDL and elevated BP or known with hypertension occurred in 94% (29/31) and 84% (26/31) respectively. The median age (IQR) of women with MetS was 35 years (IQR 30.5–40.5) and significantly older than women without MetS (31 years IQR 28–36) (P = 0.015), although 23 (74%) of the women with MetS were younger than 40 years.

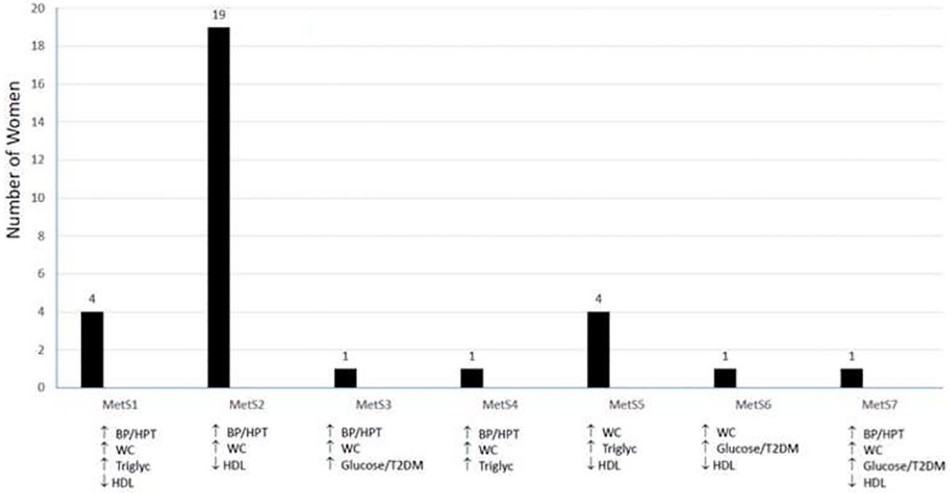

**Fig 1. Number of women in the various categories of metabolic syndrome.**

### CVD risk assessment tools

The median 5-year and 10-year CVD risk (%) by Framingham were 0.1 (IQR 0.1 to 0.3) and 0.4 (IQR 0.2 to 1.0) respectively. All women in the intervention arm were considered at low risk (<10%). The mean CVD risk (%) by DAD was 0.2±0.2 and with the exception of one woman with moderate risk (1–5%), all others were considered at low risk. All 173 women assessed by WHO/ISH were also considered at low risk (<10%).

### Obesity and self body image

Obesity was significantly more prevalent (40.8% vs 28.3%) among older women (>30 years old) than younger women (25–30 years old) (p = 0.039) (Table 3). When evaluating current satisfaction with self-body image in the intervention arm, 76.1% (140/184) of women reported to be satisfied. There was an association between satisfaction with body image and BMI (p = 0.013). Although, those who were not satisfied with body image were more likely to be in the higher BMI groups, more women who were overweight were satisfied than dissatisfied with their self-body image. Older women (>30 years) who were obese were more likely to be satisfied with their body image when compared to younger obese women (< 30 years) (75.8% vs 59.1%). Among the 79 women with both a high BMI and high WC, 52 (65.8%) were satisfied with their self body image.

## Discussion

In this cross-sectional analysis of baseline characteristics in WLHIV predominantly receiving EFV-based first line ART, we draw attention to the high prevalence of obesity (67.5%) and MetS (17.6%) among WLHIV who were mostly younger than 40. Almost all who were over-weight had a high waist circumference (WC), a combination known to be a risk for CVD. Yet, almost all of these women were considered to be low risk for CVD using Framingham, DAD and WHO/ISH tools. A significant proportion of women had decreased HDL-cholesterol levels (43.2%) and an elevated inflammatory marker, hsCRP (59.5%).

The concerning high prevalence of generalized obesity defined by elevated BMI, in our young study population, is comparable to the national prevalence in a similar age category of women independent of HIV [20]. Although this study shows similar rates to their non-HIV counterparts, this is a young cohort of WLHIV who also exhibit other risk factors for CVD associated with premature atherosclerosis [21]. Closely correlating with a high BMI, is an even higher prevalence of elevated WC, suggestive of visceral obesity. Our mean WC in women with MetS was aligned to the estimated optimal WC cut-off point of 92cm to predict the presence of at least two other components of the MetS, determined by Motala and colleagues in SA [22]. The overall mean WC of 89cm in our analysis makes a strong argument for routine WC measurements in monitoring of WLHIV.

**Table 3. Obesity and satisfaction with body image by age category.**

| Age Group | BMI < 25kg/m$^2$ | BMI 25<30 kg/m$^2$ (Overweight) | BMI ≥ 30 kg/m$^2$ (Obese) | TOTAL n |
|---|---|---|---|---|
| < 25 years | 11 (52.4%) | 5 (23.8%) | 5 (23.8%) | 21 |
| Body Image Satisfaction | - | - | 1/2 (50%) | |
| 25 < 30 years | 49 (46.2%) | 27 (25.5%) | 30 (28.3%) | 106 |
| Body Image Satisfaction | - | 14/15 (93.0%) | 13/22 (59.1%) | |
| ≥30 years | 79 (32.9%) | 63 (26.3%) | 98 (40.8%) | 240 |
| Body Image Satisfaction | - | 22/37 (81.5%) | 40/58 (75.8%) | |

South African guidelines on the management of lipid disorders in HIV-infected individuals advocate a full lipid profile at ART initiation followed by subsequent testing when treated with protease inhibitors [23]. However, lipid testing is not routinely practiced having been omitted from national ART guidelines highlighting the gaps between HIV and NCD management guidelines. Our findings are consistent with other studies that have shown that EFV adversely alters TC, LDL cholesterol, and triglycerides levels [24–26]. What stood out was a striking 43% who had low HDL levels, which advocate for routine HDL testing in primary prevention of CVD in WLHIV. Low HDL may possibly be the result of HIV and ART-related structural and functional changes [27], obesity and lifestyle practices, and is currently being explored as a potential biomarker for CVD in HIV [28].

Novel ART may improve lipid levels, and with rising HIV resistance to non-nucleoside reverse transcriptase inhibitors (NNRTI) in SA and other LMIC's, current first line ART regimens includes dolutegravir (DTG). Although DTG does not appear to cause significant change in lipids [29], findings are suggestive of DTG-associated significant weight gain, more so in women [30]. However, ongoing research is necessary to observe DTG and other integrase inhibitor's long-term effects on metabolic markers. Transition to DTG in research participants in the ISCheMiA study will shed light on this topic in future analyses.

Although elevated hsCRP was weakly associated with subclinical atherosclerosis in the absence of obesity in a United States multi-ethnic study [31], hsCRP has been shown to add value to models predicting CVD events in PLWH [32]. Furthermore, efavirenz, predominantly used in our cohort, has been associated with a higher increase in hsCRP compared to other ART [33]. The significantly high hsCRP levels in our study warrants long-term monitoring of CVD risk, but disentangling obesity and hsCRP levels remains a challenge. A recent recommendation by WHO is the conduct of a CVD risk assessment in PLWH, which has recently been incorporated into the latest SA HIV management guidelines. This practice is not yet widely implemented. The CARDIA study showed young adults with detectable risk factors at baseline, were up to 3 times as likely to have coronary artery plaque calcification suggesting that intervention when risk factor levels reach management guideline thresholds may be too late in preventing CVD [34].

While it is recognised that it would be impossible to accurately estimate risk in all South African subpopulations with a single data set, the Fr was the first choice in our study, having been validated in white and black populations in the USA and are transportable to other culturally diverse populations [17]. Nevertheless, these risk tables are likely to underestimate risk in South African black and Indian patients, and people with HIV, apparent in our study population who had almost 18% prevalence of MetS yet low CVD risk scores. Hence the use of DAD (for HIV), and WHO/ISH which is specific to the South African region. The DAD does not account for low income settings, and WHO/ISH does not incorporate HIV and young adults. Clinical discretion is strongly advised when selecting an appropriate risk assessment tool.

Finally, women in our study population appear to be content with being overweight. Similarly, Malawian women with HIV preferred to be overweight as it was associated with the ability to breast-feed [13]. Unhealthy diet and lack of exercise are targets for our study intervention, however interim analyses suggest that women are not adhering to lifestyle modification advice after 6 months follow-up. Further exploration of body self-image and body-satisfaction is necessary where we are seeing a move in focus of body image from HIV lipodystrophy to obesity.

Health care provider and patient awareness of the heightened CVD risk as well as the potential influence of self body image in WLHIV may facilitate a shift in the typical lifestyle modification advice approach. This change in practice could positively impact on the number

of disability-associated life-years (DALYs) related to CVD in PLWH in Sub-Saharan Africa, which currently holds a third of the global annual DALYs [35].

## Limitations

Co-enrolment into intervention arm from the ongoing observational PROMOTE study may introduce selection bias, however women receive care from standard of care providers and no prior targeted CVD screening was performed. When comparing the distribution of the CVD risk factors determined as per standard of care presented in Table 1 between the two study arms, only BMI and CD4 differed between control and intervention. The control were higher in BMI and had lower CD4 counts.

## Conclusion

Although traditional CVD risk assessments yielded low risk scores in this relatively young cohort of WLHIV, there was a high prevalence of metabolic syndrome, obesity, abnormal HDL levels and elevated markers of inflammation, combined with self-reported unhealthy lifestyle practices and satisfaction with being overweight. These findings, compounded by the additional known and emerging effects of HIV and ART, highlight the need for a pro-active integrated differentiated care approach to the primary prevention of CVD in young WLHIV, preferably initiated at first point of contact at HIV treatment centres. Opportunistic health promotion in ARV clinics may be a good starting point to curb overall NCD incidence in LMICs.

## Supporting information

**S1 File.**
(DOCX)

**S2 File.**
(PDF)

**S3 File.**
(XLSX)

## Acknowledgments

The authors would like to acknowledge the research participants and PROMOTE study team, as well as contributions from research assistants Zinhle Shazi and Nonhlanhla Silindana, statisticians Nonhlanhla Yende-Zuma and Tonya Esterhuizen and the CAPRISA Data Management Centre.

## Author Contributions

**Conceptualization:** Sherika Hanley.

**Data curation:** Sherika Hanley, Dhayendre Moodley.

**Formal analysis:** Sherika Hanley, Dhayendre Moodley, Mergan Naidoo.

**Funding acquisition:** Sherika Hanley.

**Investigation:** Sherika Hanley.

**Methodology:** Sherika Hanley, Dhayendre Moodley, Mergan Naidoo.

**Project administration:** Sherika Hanley, Dhayendre Moodley.

**Resources:** Sherika Hanley, Dhayendre Moodley.

**Software:** Dhayendre Moodley.

**Supervision:** Dhayendre Moodley, Mergan Naidoo.

**Writing – original draft:** Sherika Hanley, Dhayendre Moodley, Mergan Naidoo.

**Writing – review & editing:** Sherika Hanley, Dhayendre Moodley, Mergan Naidoo.

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
