## [Decision Letter · Decision Letter 0]

17 Nov 2020

PONE-D-20-27919

Obesity in young South African women living with HIV: a cross-sectional analysis of risk factors for cardiovascular disease

PLOS ONE

Dear Dr. Sherika Hanley,

Thank you for submitting your manuscript to PLOS ONE. After careful consideration, we feel that it has merit but does not fully meet PLOS ONE’s publication criteria as it currently stands. Therefore, we invite you to submit a revised version of the manuscript that addresses the points raised during the review process.

ACADEMIC EDITOR:

As highlighetd by one reviewer, some issues on body size, particularly WC, need clarification and missing data, eg on Lp(a) should be justified.

We look forward to receiving your revised manuscript.

Kind regards,

Massimiliano Ruscica, Ph.D.

Academic Editor

PLOS ONE

Journal Requirements:

2. In addition, alongside your ethics statement we would ask that you include your trial registration details, namely where the trial was registered and the registration number( PACTR201808524461224).

3. Please include your tables as part of your main manuscript and remove the individual files. Please note that supplementary tables (should remain/ be uploaded) as separate "supporting information" files.

Additional Editor Comments (if provided):

As suggested by one of the reviewer,

Reviewers' comments:

Reviewer's Responses to Questions

**Comments to the Author**

1. Is the manuscript technically sound, and do the data support the conclusions?

Reviewer #1: Yes

Reviewer #2: Yes

2. Has the statistical analysis been performed appropriately and rigorously? 

Reviewer #1: Yes

Reviewer #2: Yes

3. Have the authors made all data underlying the findings in their manuscript fully available?

Reviewer #1: Yes

Reviewer #2: Yes

4. Is the manuscript presented in an intelligible fashion and written in standard English?

Reviewer #1: Yes

Reviewer #2: No

5. Review Comments to the Author

Reviewer #1: The prevalence of risk factors for CVD, their interrelationship and resultant CVD risk scores were investigated cross-sectionally in a cohort of young South African women undergoing treatment for HIV.

While it is clearly valuable to report on the degree of CVD risk and specific risk factors in particular populations, such as here young women with HIV, it is unclear to me what the results in this relatively small cohort add to our knowledge. Does it contribute anything new concerning prevalence of obesity, other than to confirm that young South African women with HIV show similar rates to their non-HIV counterparts?

A second main point: the account of analyses of the relationships among the various risk factors is poorly structured and difficult to grasp. These were all baseline variables, so the motivation for regression analyses which assign one variable as dependent and others as independent is unclear to me. Since all variables appear to have been dichotomised for these analyses, would in not be simpler to present a matrix of odds ratios between each pair of factors, or alternatively just report those ORs which were significant and state that all others were not significant?

Finally, it is stated that all participants were ‘low risk’ according to Framingham score (please state exactly which score and provide a reference). Indeed, the mean risk is just 0.3%, whereas the threshold for moderate risk lies at 10%. This seems to contradict the high prevalence of obesity and other CVD risk factors, as well as the fairly high prevalence of metabolic syndrome, and needs to be explained.

Minor points:

1. Please explain how the control cohort was matched for age and ART duration with the intervention cohort (line 91).

2. The method of calculating confidence intervals for percentages in Tables 1 and 2 should be given (line 137).

3. More detail on regression methods (including which variables were dependent, and whether stepwise) are needed, if regression is retained in the manuscript (lines 135…).

4. Line 141: sentence unclear – relationship between perception and BMI?

5. Table 2: why are parity mean values integers?

6. Line 199: mean Framingham scores were 0.3 ±0.6 and 0.9±1.2. Are the ± values standard errors or confidence intervals? These ranges include negative values and thus seem to be implausible, maybe due to skewness of the distribution.

7. Selection bias in the intervention cohort (line 346) could be investigated by comparing the distribution of the variables and factors presented in Table 1 between the two cohorts, intervention and control.

8. Fig 1 is difficult to grasp visually; in particular the colour bars belonging to a combination are not always adjacent. Might be easier if the bars (one bar per combination) were labelled directly with the combination of factors, rather than using colours.

9. Fig. 2: might be more useful to calculate percentages per BMI category rather than per satisfaction group.

Reviewer #2: The paper by Hanley et al provides a potentially important contribution on the cardiovascular risk in young South African women with AIDS and overweight.

The Authors give a lot of information in the Supplements, most of which (budget, change in protocol etc) are of no interest to readers and should be left out.

There are instead a few problems in the text/Methods.

Introduction

The mechanism of raised CV risk by immune activation is one of the many. The paper by SA Authors is just an opinion paper and the issue should be better supported or left out.

There is repeated emphasis on self (not selff) preference for overweight in these individuals, but the supporting criteria are very weak (is it just a simple question: do you like yourself: yes or no?) and really has little meaning for non SA readers.

In the Methods one does not really understand how recruitment occurs. From what I understand ART is the major criterion. What are the controls? Nearest place?

Waist circumference is measured at the iliac crest? This is certainly wrong. It is either midway between the iliac crest and the lowest rib or at the umbilical level. The figures I see are way too low for the type of measurement they carried out.

p.7 probably Becton Dickinson

MetSyn is by criteria used in low income nations, particularly for WC. Since most participants had WC above 80 cm, the Authors should either justify this or recalculate data based on European/US parameters.

In my view using the Framingham risk prediction makes little sense (essentially all participants were at very low levels). Please restrict yourselves to the DAD equation.

In the Results it is inappropriate to use brand names for medicines: some have become generic. TRUVADA is emtricitabine/tenofovir etc.

Coming to the body variables, the Authors should list the WHt ratio a very selective and sensitive parameter for the MetSyn (Pavanello et al. , Influence of body variables in the development of metabolic syndrome-A long term follow-up study PLoS One. 2018 Feb 12;13(2):e0192751) . They have the data, the advantage is that this is a single marker. The color figures are horrible, Fig. 1 could be more understandable with clear markers, Fig.2 on self-image, is worthless and should be left out.

I have already pointed out the problems of WC. Assessment of risk by the FRS should be left out, whereas some more information should be provided on DAD.

What is Lipogram? Essentialy you measure lipids: cholesterol,TG, HDL-C, no data on Lp(a)?

Finally the self-body image must be left out.

In the Discussion, overly lengthy, they present a summary of findings, of no value, and indicate that a WC of 91 cm is probably the real threshold for MetSyn. ART is of little concern except probably for dolutegravir. If so, why using this drug?

The small presence of albuminuria is not worth discussing, whereas the lipid abnormalities are examined very briefly. Why low HDL-C in so many individuals? Again, these are the issues that merit discussion, not self-body image. Discussion should be cut to not more than one half.

6. PLOS authors have the option to publish the peer review history of their article (what does this mean?). If published, this will include your full peer review and any attached files.

Reviewer #1: **Yes: **Jeremy Franklin

Reviewer #2: No

---

## [Author Response · Author response to Decision Letter 0]

18 Feb 2021

Response to Reviewers

ACADEMIC EDITOR comments:

As highlighted by one reviewer, some issues on body size, particularly WC, need clarification and missing data, eg on Lp(a) should be justified.

Thank you for the opportunity to revise my manuscript. 

The WC and missing Lp(a) has been clarified below. 

The method of WC measurement has been described: Waist circumference (WC) was measured by snugly placing a measuring tape in a horizontal plane around the abdomen immediately above the iliac crest at the level of the umbilicus, at the end of expiration.

According to the South African Dyslipidaemia Guidelines, additional Lp(a) measurement may be considered for reclassification of subjects falling on a borderline between moderate and high risk. Lp(a) screening can also be considered for selected individuals at high CVD risk, including those with premature CVD, FH, a family history of premature CVD and/or elevated Lp(a), recurrent CVD despite optimal lipid-lowering treatment, and risk ≥15% on the 10-year Framingham risk tables. The author’s intention was to assess baseline CVD risk and determine additional tests to be conducted during the course of the study.

Reviewer #1: 

1. The prevalence of risk factors for CVD, their interrelationship and resultant CVD risk scores were investigated cross-sectionally in a cohort of young South African women undergoing treatment for HIV.

While it is clearly valuable to report on the degree of CVD risk and specific risk factors in particular populations, such as here young women with HIV, it is unclear to me what the results in this relatively small cohort add to our knowledge. Does it contribute anything new concerning prevalence of obesity, other than to confirm that young South African women with HIV show similar rates to their non-HIV counterparts?

Thank you. We recognise that the mere prevalence of obesity does not add to the readers’ knowledge. We do however want to remind the reader that the high prevalence of obesity in this fairly young cohort of WLHV together with other risk factors for CVD associated with premature mortality, and in keeping with the National Non-Communicable Disease Strategic Plan for South Africa, we should focus attention on primary prevention and management of these risk factors as part of our integrated plan to care for WLHIV. 

2. A second main point: the account of analyses of the relationships among the various risk factors is poorly structured and difficult to grasp. These were all baseline variables, so the motivation for regression analyses which assign one variable as dependent and others as independent is unclear to me. Since all variables appear to have been dichotomised for these analyses, would in not be simpler to present a matrix of odds ratios between each pair of factors, or alternatively just report those ORs which were significant and state that all others were not significant?

Thank you for raising this point. We have corrected the analyses.

3. Finally, it is stated that all participants were ‘low risk’ according to Framingham score (please state exactly which score and provide a reference). Indeed, the mean risk is just 0.3%, whereas the threshold for moderate risk lies at 10%. This seems to contradict the high prevalence of obesity and other CVD risk factors, as well as the fairly high prevalence of metabolic syndrome, and needs to be explained.

Centre of Excellence for Health, Immunity and Infections (CHIP), University of Copenhagen Risk assessment tool system (RATS)-The Framingham algorithm estimates the risk of developing a cardiovascular disease within the next 5 years (modified to be compared with the D:A:D CVD 5 year risk score) and next 10 years (original Framingham risk score). The Framingham model is valid for individuals aged 30 to 75. Required information: Gender, age, smoking status, diabetes (diagnosis or on antidiabetic treatment), systolic BP, antihypertensive treatment, total cholesterol, HDL. 

Reference: D'Agostino RB Sr, Vasan RS, Pencina MJ, Wolf PA, Cobain M, Massaro JM, Kannel WB. General cardiovascular risk profile for use in primary care: the Framingham Heart Study. Circulation. 2008 Feb 12;117(6):743-53. doi: 10.1161/CIRCULATIONAHA.107.699579. Epub 2008 Jan 22. PMID: 18212285.

The low risk scores generated by Framingham highlight the limitations of current CVD risk assessment tools in young people with HIV. The young age of the study population, and the exclusion of obesity and HIV risk factors, have resulted in low scoring.

4. Minor points:

1. Please explain how the control cohort was matched for age and ART duration with the intervention cohort (line 91). 

Method of selection was via convenience sampling. 

Intervention group: All 238 women enrolled into the parent PEPFAR PROMise Ongoing Treatment Evaluation (PROMOTE) observational study at the CAPRISA Research Clinic in Umlazi, are living with HIV and are between 18 and 49 years of age, and on ART for more than 1 year. This study has been implemented to provide long-term follow-up data on safety outcomes of widespread use of combination antiretrovirals (cART) among an already well-characterized cohort of HIV infected mothers and their children who previously enrolled in the multi-site PROMISE study. Women were briefed at their next PROMOTE study visit and the first 186 interested candidates meeting all eligibility criteria were co-enrolled into the Intervention arm of the ISCheMiA study. 

Control group: The Tier data base was used to select a data base of all women with HIV aged between 18-49 years, receiving ART for more than 1 year at the nearest Umlazi Gateway PHC. Scheduled clinic visits at similar time points to the anticipated clinic visits in the intervention group were used to establish a list of potentially eligible women. Following a matched pool of data, the first 186 women fulfilling the inclusion criteria, who attended the clinic for their next appointment and who consent to study participation were enrolled. 

2. The method of calculating confidence intervals for percentages in Tables 1 and 2 should be given (line 137).

Using Epiinfo 7.0, calculation of the confidence interval is based on a mathematical relation between the binomial distribution and F distribution (Fisher & Yates, 1963; Zar, 1996, p. 524).

3. More detail on regression methods (including which variables were dependent, and whether stepwise) are needed, if regression is retained in the manuscript (lines 135…).

Regression analyses has been removed.

4. Line 141: sentence unclear – relationship between perception and BMI?

The sentence has been restructured.

5. Table 2: why are parity mean values integers?

Parity is corrected to median and IQR.

6. Line 199: mean Framingham scores were 0.3 ±0.6 and 0.9±1.2. Are the ± values standard errors or confidence intervals? These ranges include negative values and thus seem to be implausible, maybe due to skewness of the distribution.

Since the Framingham scores are very skew, median and IQr has been reported: 5 yr median = 0.1 (IQR 0.1 to 0.3) and 10 year median = 0.4 (IQR 0.2 to 1.0)

7. Selection bias in the intervention cohort (line 346) could be investigated by comparing the distribution of the variables and factors presented in Table 1 between the two cohorts, intervention and control.

Only BMI and CD4 differed between control and intervention. The control were higher in BMI and had lower CD4 counts.

8. Fig 1 is difficult to grasp visually; in particular the colour bars belonging to a combination are not always adjacent. Might be easier if the bars (one bar per combination) were labelled directly with the combination of factors, rather than using colours.

Figure 1 has been revised.

9. Fig. 2: might be more useful to calculate percentages per BMI category rather than per satisfaction group.

Figure 2 has been replaced by Table 3.

 

Reviewer #2: The paper by Hanley et al provides a potentially important contribution on the cardiovascular risk in young South African women with AIDS and overweight.

Thank you

1.The Authors give a lot of information in the Supplements, most of which (budget, change in protocol etc) are of no interest to readers and should be left out.

Information left out

2.There are instead a few problems in the text/Methods.

Introduction

The mechanism of raised CV risk by immune activation is one of the many. The paper by SA Authors is just an opinion paper and the issue should be better supported or left out.

Further reference added in support

3.There is repeated emphasis on self (not selff) preference for overweight in these individuals, but the supporting criteria are very weak (is it just a simple question: do you like yourself: yes or no?) and really has little meaning for non SA readers.

The author acknowledges the weak data collection tool for the image analyses (It was a simple yes or no question as to whether participants were currently satisfied with their current body image). The authors have been working on an improved collection tool for future qualitative analysis. The authors do believe that there are multifactorial causes for obesity in young women and would like to further explore, create interest and expand on the typical lifestyle modification advice that clinicians have been using for years.

4.In the Methods one does not really understand how recruitment occurs. From what I understand ART is the major criterion. What are the controls? Nearest place?

Method of selection was via convenience sampling. 

Intervention group: All 238 women enrolled into the parent PEPFAR PROMise Ongoing Treatment Evaluation (PROMOTE) observational study at the CAPRISA Research Clinic in Umlazi, are living with HIV and are between 18 and 49 years of age, and on ART for more than 1 year. This study has been implemented to provide long-term follow-up data on safety outcomes of widespread use of combination antiretrovirals (cART) among an already well-characterized cohort of HIV infected mothers and their children who previously enrolled in the multi-site PROMISE study. Women were briefed at their next PROMOTE study visit and the first 186 interested candidates meeting all eligibility criteria were co-enrolled into the Intervention arm of the ISCheMiA study. 

Control group: The Tier data base was used to select a data base of all women with HIV aged between 18-49 years, receiving ART for more than 1 year at the nearest PHC, Umlazi Gateway PHC. Scheduled clinic visits at similar time points to the anticipated clinic visits in the intervention group were used to establish a list of potentially eligible women. Following a matched pool of data, the first 186 women fulfilling the inclusion criteria, who attended the clinic for their next appointment and who consent to study participation were enrolled.

5.Waist circumference is measured at the iliac crest? This is certainly wrong. It is either midway between the iliac crest and the lowest rib or at the umbilical level. The figures I see are way too low for the type of measurement they carried out.

Waist circumference (WC) was measured by snugly placing a measuring tape in a horizontal plane around the abdomen immediately above the iliac crest at the level of the umbilicus, at the end of expiration

5. p.7 probably Becton Dickinson

Thank you, spelling error correction from Dickenson to Dickinson

6. MetSyn is by criteria used in low income nations, particularly for WC. Since most participants had WC above 80 cm, the Authors should either justify this or recalculate data based on European/US parameters.

The authors acknowledges the vast ethnic diversity in South Africa, however for consistency opted to restrict MetSyn diagnostic parameters to South African guidelines. The South African Heart Association (SA Heart) and the Lipid and Atherosclerosis Society of Southern Africa (LASSA) have recommended the JIS guidelines. This has been further clarified in the method section of the manuscript. Long-term prospective studies are required to reach more reliable waist circumference cut points for different ethnic groups, particularly for women.

7. In my view using the Framingham risk prediction makes little sense (essentially all participants were at very low levels). Please restrict yourselves to the DAD equation. 

While it is recognised that it would be impossible to accurately estimate risk in all South African subpopulations with a single data set, the Adult Treatment Panel (ATP) III Framingham risk tables which provide an estimate of the 10-year risk of CHD, have been validated in white and black populations in the USA and are transportable to other culturally diverse populations. Consequently, we considered this approach to be more appropriate for South Africa. Nevertheless, these risk tables are likely to underestimate risk in South African black and Indian patients, and patients with HIV. Hence the use of DAD (for HIV), and WHO/ISH which is specific to the South African region. The DAD could not be the single means of assessing CVD risk as it has been developed in high income settings and do not account for low income settings

8. In the Results it is inappropriate to use brand names for medicines: some have become generic. TRUVADA is emtricitabine/tenofovir etc.

Thank you for pointing this out. All brand names in the results table have been removed.

9.Coming to the body variables, the Authors should list the WHt ratio a very selective and sensitive parameter for the MetSyn (Pavanello et al. , Influence of body variables in the development of metabolic syndrome-A long term follow-up study PLoS One. 2018 Feb 12;13(2):e0192751) . They have the data, the advantage is that this is a single marker. The color figures are horrible, Fig. 1 could be more understandable with clear markers, Fig.2 on self-image, is worthless and should be left out.

Thank you for this useful recommendation. As mentioned above, for consistency the authors have opted to restrict MetSyn diagnostic parameters to South African recommendations.

Figure 1 has been revised. Figure 2 has been replaced by Table 3.

10. I have already pointed out the problems of WC. Assessment of risk by the FRS should be left out, whereas some more information should be provided on DAD. 

Explanation is provided above.

11. What is Lipogram? Essentialy you measure lipids: cholesterol,TG, HDL-C, no data on Lp(a)?

Finally the self-body image must be left out.

Thank you for raising this. The term lipogram has been replaced by lipid profile.

A lipogram and lipid profile is used interchangeably in South Africa, and refers to a standard set of tests: total cholesterol, LDL cholesterol, HDL cholesterol and triglycerides. According to the South African Dyslipidaemia Guidelines, additional Lp(a) measurement may be considered for reclassification of subjects falling on a borderline between moderate and high risk. Lp(a) screening can also be considered for selected individuals at high CVD risk, including those with premature CVD, FH, a family history of premature CVD and/or elevated Lp(a), recurrent CVD despite optimal lipid-lowering treatment, and risk ≥15% on the 10-year Framingham risk tables. The author’s intention was to assess baseline CVD risk and determine additional tests to be conducted during the course of the study.

12. In the Discussion, overly lengthy, they present a summary of findings, of no value, and indicate that a WC of 91 cm is probably the real threshold for MetSyn. ART is of little concern except probably for dolutegravir. If so, why using this drug?

Discussion has been revised/shortened.

13. The small presence of albuminuria is not worth discussing, whereas the lipid abnormalities are examined very briefly. Why low HDL-C in so many individuals? Again, these are the issues that merit discussion, not self-body image. Discussion should be cut to not more than one half.

Discussion on albuminuria and other sections have been left out/reduced.

Use of Dolutegravir and Low HDL elaborated on.

The brief discussion on self image may allow for exploration into different strategies when providing lifestyle modification advice to women

---

## [Decision Letter · Decision Letter 1]

15 Mar 2021

PONE-D-20-27919R1

Obesity in young South African women living with HIV: a cross-sectional analysis of risk factors for cardiovascular disease

PLOS ONE

Dear Dr. %Sherika Hanley%,

Thank you for submitting your manuscript to PLOS ONE. After careful consideration, we feel that it has merit but does not fully meet PLOS ONE’s publication criteria as it currently stands. Therefore, we invite you to submit a revised version of the manuscript that addresses the points raised during the review process.

We look forward to receiving your revised manuscript.

Kind regards,

Massimiliano Ruscica, Ph.D.

Academic Editor

PLOS ONE

Journal Requirements:

Reviewers' comments:

Reviewer's Responses to Questions

**Comments to the Author**

1. If the authors have adequately addressed your comments raised in a previous round of review and you feel that this manuscript is now acceptable for publication, you may indicate that here to bypass the “Comments to the Author” section, enter your conflict of interest statement in the “Confidential to Editor” section, and submit your "Accept" recommendation.

Reviewer #1: (No Response)

Reviewer #2: All comments have been addressed

2. Is the manuscript technically sound, and do the data support the conclusions?

Reviewer #1: Yes

Reviewer #2: Yes

3. Has the statistical analysis been performed appropriately and rigorously? 

Reviewer #1: Yes

Reviewer #2: Yes

4. Have the authors made all data underlying the findings in their manuscript fully available?

Reviewer #1: Yes

Reviewer #2: Yes

5. Is the manuscript presented in an intelligible fashion and written in standard English?

Reviewer #1: Yes

Reviewer #2: Yes

6. Review Comments to the Author

Reviewer #1: The problematic regression analyses have been omitted from the revised version. Although I believe they could have been replaced by more meaningful methods, this is not essential to the manuscript.

The authors have responded appropriately to the comments I made; only the following still need to be attended to:

1. Please add details of matching in the control group, as requested by both reviewers. The text in the reply to reviewers would form a good basis.

2. State the method used to calculate confidence intervals for proportions. The statement in the reply about Fisher and Yates and the F distributiion is rather too vague.

3. Minor point 7: on the topic of selection bias, under 'Limitations' is would be informative to briefly describe the comparison of characteristics in the intervention and control groups.

4. Finally, the special relevance of the findings for women with HIV, as opposed to other young women, including the consequences for health care, should perhaps be more clearly stated in the discussion.

Reviewer #2: You have done an adequate job. The differences with other populations are there, but i believe you cannot do anything about it

7. PLOS authors have the option to publish the peer review history of their article (what does this mean?). If published, this will include your full peer review and any attached files.

Reviewer #1: **Yes: **Jeremy Franklin

Reviewer #2: No

---

## [Author Response · Author response to Decision Letter 1]

5 Apr 2021

Thank you for the opportunity to submit a revised version of my manuscript entitled “Obesity in young South African women living with HIV: a cross-sectional analysis of risk factors for cardiovascular disease”.

Please see responses to each point raised by the academic editor and reviewer below.

Journal Requirements:

Response: Reference list has been reviewed and is complete and correct. No papers have been retracted.

The most recent UNAIDS data sheet has been referenced in the introduction section. 

New reference no. 35. Shah ASV, Stelzle D, Lee KK, et al. Global Burden of Atherosclerotic Cardiovascular Disease in People Living With HIV: Systematic Review and Meta-Analysis. Circulation. 2018;138(11):1100-1112.

Reviewer #1: 

Query 1. Please add details of matching in the control group, as requested by both reviewers. The text in the reply to reviewers would form a good basis.

Response 1. Thank you. The details of matching in the control group have been included in the text of the manuscript.

Query 2. State the method used to calculate confidence intervals for proportions. The statement in the reply about Fisher and Yates and the F distribution is rather too vague.

Response 2. 95% confidence intervals were derived from using Epiinfo 7.0 and were either calculated as a Wilson interval or an Exact interval where appropriate. Wilson 95% Confidence Interval was calculated for Systolic BP, CD4, Age, BMI, ART and ART Duration Groups. Exact 95% Confidence Interval was calculated for all other categories. This has been added in the text under statistical analyses.

Query 3. Minor point 7: on the topic of selection bias, under 'Limitations' it would be informative to briefly describe the comparison of characteristics in the intervention and control groups. 

Response 3. Thank you. This information has been added under the Limitations section.

Query 4. Finally, the special relevance of the findings for women with HIV, as opposed to other young women, including the consequences for health care, should perhaps be more clearly stated in the discussion.

Response 4. The author has made additional reference to disability-associated life-years from CVD in persons with HIV in the discussion without significantly altering the length of the discussion, and has placed emphasis on women with HIV in the conclusion.

---

## [Decision Letter · Decision Letter 2]

22 Jul 2021

Obesity in young South African women living with HIV: a cross-sectional analysis of risk factors for cardiovascular disease

PONE-D-20-27919R2

Dear Dr. Hanley,

We’re pleased to inform you that your manuscript has been judged scientifically suitable for publication and will be formally accepted for publication once it meets all outstanding technical requirements.

Kind regards,

Massimiliano Ruscica, Ph.D.

Academic Editor

PLOS ONE

Additional Staff Editor Comments:

Alongside your ethics statement, please include your trial registration details, namely where the trial was registered and the registration number

Reviewers' comments:

Reviewer's Responses to Questions

**Comments to the Author**

1. If the authors have adequately addressed your comments raised in a previous round of review and you feel that this manuscript is now acceptable for publication, you may indicate that here to bypass the “Comments to the Author” section, enter your conflict of interest statement in the “Confidential to Editor” section, and submit your "Accept" recommendation.

Reviewer #1: All comments have been addressed

2. Is the manuscript technically sound, and do the data support the conclusions?

Reviewer #1: (No Response)

3. Has the statistical analysis been performed appropriately and rigorously? 

Reviewer #1: (No Response)

4. Have the authors made all data underlying the findings in their manuscript fully available?

Reviewer #1: (No Response)

5. Is the manuscript presented in an intelligible fashion and written in standard English?

Reviewer #1: (No Response)

6. Review Comments to the Author

Reviewer #1: (No Response)

7. PLOS authors have the option to publish the peer review history of their article (what does this mean?). If published, this will include your full peer review and any attached files.

Reviewer #1: **Yes: **Jeremy Franklin

---

## [Editor Report · Acceptance letter]

5 Nov 2021

PONE-D-20-27919R2 

Obesity in young South African women living with HIV: a cross-sectional analysis of risk factors for cardiovascular disease 

Dear Dr. Hanley:

I'm pleased to inform you that your manuscript has been deemed suitable for publication in PLOS ONE. Congratulations! Your manuscript is now with our production department. 

Kind regards, 

on behalf of

Dr. Massimiliano Ruscica 

Academic Editor

PLOS ONE